# Endoplasmic Reticulum Calcium Signaling in Hippocampal Neurons

**DOI:** 10.3390/biom14121617

**Published:** 2024-12-18

**Authors:** Vyacheslav M. Shkryl

**Affiliations:** Department of Biophysics of Ion Channels, Bogomoletz Institute of Physiology, NAS of Ukraine, 01024 Kyiv, Ukraine; slava@biph.kiev.ua

**Keywords:** endoplasmic reticulum, ryanodine receptor, inositol 1,4,5-trisphosphate receptor, calcium signaling, neurons

## Abstract

The endoplasmic reticulum (ER) is a key organelle in cellular homeostasis, regulating calcium levels and coordinating protein synthesis and folding. In neurons, the ER forms interconnected sheets and tubules that facilitate the propagation of calcium-based signals. Calcium plays a central role in the modulation and regulation of numerous functions in excitable cells. It is a versatile signaling molecule that influences neurotransmitter release, muscle contraction, gene expression, and cell survival. This review focuses on the intricate dynamics of calcium signaling in hippocampal neurons, with particular emphasis on the activation of voltage-gated and ionotropic glutamate receptors in the plasma membrane and ryanodine and inositol 1,4,5-trisphosphate receptors in the ER. These channels and receptors are involved in the generation and transmission of electrical signals and the modulation of calcium concentrations within the neuronal network. By analyzing calcium fluctuations in neurons and the associated calcium handling mechanisms at the ER, mitochondria, endo-lysosome and cytosol, we can gain a deeper understanding of the mechanistic pathways underlying neuronal interactions and information transfer.

## 1. Introduction

A neuron is a specialized excitable cell that acts as the basic unit of the nervous system. Its primary functions include conveying information, transmitting signals, and allowing communication between organs to the central nervous system (CNS). As part of the CNS, the hippocampus is essential for spatial navigation and the formation of new declarative and episodic memories. As a brain region with high plasticity, the hippocampus is vulnerable to a variety of events, including hypoxia/ischemia, epilepsy, depression, stress, aging, and Alzheimer’s disease [1]. Hippocampal atrophy is a permanent feature of Alzheimer’s disease, leading to dissociation between the hippocampus and the cerebral cortex [2]. Hippocampal neurons are well suited for analysis of calcium signaling because they exhibit complex intracellular calcium dynamics mediated by pre-membrane channels and SR receptors in the soma of the neuron, and a dendritic architecture with spines facilitates localized calcium signaling that can be precisely regulated.

Signal processing in neurons involves short electrical signals across the membrane that are converted into biochemical responses, leading to changes in the structural and functional states of neurons over time [3]. Action potentials activate voltage-gated channels that could increase calcium concentration within axons and presynaptic terminals, and these signals can backpropagate into the dendrites [4]. Calcium ions (Ca^2+^) are a secondary mediator that regulates or controls many physiological pathways in excitable cells, such as excitation, muscle contraction, mediator release, synaptic plasticity, secretion, gene transcription, apoptosis, etc. [5,6,7]. In the resting state, most neurons typically exhibit a relatively low intracellular concentration of free calcium, ranging from 50 to 100 nM. This concentration can surge to as high as 10 μM during periods of intense electrical activity [6]. Additionally, calcium signals resulting from the release of this ion from intracellular stores have been observed in neurons experiencing various physiological stimuli [8]. The neuronal plasma membrane, equipped with various channels, receptors, and pumps, facilitates the transmission of electrical signals.

In the majority of CNS neurons, changes in free intracellular calcium concentration ([Ca^2+^]_i_) occur due to the activation of voltage-gated calcium channels (VGCCs) or ligand-gated channels, including neuronal glutamate receptors [9,10,11]. Calcium is subsequently buffered by calcium-binding proteins, including calmodulin, calreticulin, parvalbumin, and calbindin-D28k. The ER membrane also contributes to the propagation of calcium-based signals, primarily along the dendritic length and soma, moving from the periphery to the center of the cell. Similar to the attenuation of electrical potentials, the flux of calcium through ER receptors results in the passive diffusion of calcium to nearby sites [12,13,14]. Inositol 1,4,5-trisphosphate receptors (IP_3_Rs) and ryanodine receptors (RyRs) are located in the ER and generate Ca^2+^ signals in response to neuronal activity. The ER extends throughout the neuronal structure, allowing intracellular Ca^2+^ release in dendrites, somatic regions, and presynaptic boutons [15]. The major channels and receptors involved in Ca^2+^ signaling in neurons are shown in Figure 1.

The ER plays a crucial role in maintaining cellular homeostasis by regulating calcium levels and coordinating protein synthesis and folding. Immunohistochemical and electron microscopic investigations have confirmed the existence of the ER in neuronal cell bodies, dendrites, and axons. Calcium is released from the ER through IP_3_Rs and RyRs. IP_3_Rs are activated by neurotransmitters, while RyRs are activated by an increase in cytosolic Ca^2+^ concentration. The ER Ca^2+^ depletion initiates a cascade leading to store-operated Ca^2+^ entry, which could also be modulated by IP_3_Rs in neurons. The binding of IP_3_ to these receptors facilitates the association between STIM1 and Orai1 independently of their Ca^2+^-releasing function. The release of calcium from the ER in neurons is pivotal for synaptic plasticity and the propagation of calcium waves. These waves, driven by CICR, can originate in dendrites and spread throughout the neuron, facilitating communication between synapses and the nucleus. The geometry of the ER and the density of RyRs influence the generation and propagation of these waves. Dendritic spines, crucial for signal integration, contain VGCCs that facilitate calcium influx and backpropagation of action potentials. Calcium release from the ER through intracellular calcium release channels, such as IP_3_Rs and RyRs, contributes to spine-specific calcium signaling and plasticity. The ER within the dendritic spines, known as the spine apparatus, plays a role in synaptic plasticity by enhancing calcium signaling. Mitochondrial calcium uptake through the mitochondrial calcium uniporter influences cellular energy metabolism and can lead to a change in calcium signaling. Mitochondrial Ca^2+^ waves triggered by elevated cytosolic Ca^2+^ help regulate cellular energy metabolism and prevent overloading. Endo-lysosomes (ELs) are acidic organelles that participate in calcium signaling and are involved in endocytosis, recycling, and degradation. Calcium release from endo-lysosomes can increase the ER calcium load and initiate vesicular fusion.

A comprehensive understanding of Ca^2+^ signaling in hippocampal neurons is of paramount importance, as it constitutes an indispensable component of neurotransmitter release, synaptic plasticity, gene expression, transcription, neuron excitability and action potential propagation. Abnormalities in Ca^2+^ dynamics have been linked to some neurological disorders, including Alzheimer’s disease, Huntington’s disease, and epilepsy [16,17,18]. By examining specific mechanisms, such as the role of the ER, IP_3_, and RyRs, the involvement of calcium handling in dendrites, spines, pre-membrane regions, and inside the soma of neurons, enhances our fundamental understanding of these processes, which could contribute to the development of new therapeutic approaches to treat neurological diseases.

## 2. The Endoplasmic Reticulum and Neuron Ca^2+^ Signaling

The ER is a vital cellular organelle that plays a central role in maintaining homeostasis by regulating calcium levels and coordinating key processes such as protein and lipid synthesis, folding and storage [19]. Uniquely, the ER is a widespread organelle characterized by a continuous lumen that serves as a Ca^2+^ reservoir, sustaining free Ca^2+^ concentrations of 500 μM or more, which is maintained by active Ca^2+^ transport via Ca^2+^-ATPases and the presence of numerous calcium-binding proteins [10,16].

In neurons, the ER, which has a surface area far greater than the plasma membrane, forms interconnected sheets within the soma that extend to dendrites and dendritic spines and the cisternal organelle in the axon initial segment [20]. In some regions, it may also take on a tubular shape [21,22]. The axonal ER is elongated tubules with a diameter of ~25 nm [23]. The ER sheets in the neuronal soma and dendrites are continuous and undergo dynamic membrane exchange, a process essential for maintaining long-term Ca^2+^ homeostasis in dendrites [24,25,26]. Areas of the ER located adjacent to the plasma membrane (PM) create specialized membrane domains referred to as ER-PM junctions, located less than 10 nm from the PM [27,28]. In both neurons and non-neuronal cells, L-type Ca^2+^ channels cluster at the PM, where they form part of the protein complexes that link L-type Ca^2+^ channel-mediated Ca^2+^ entry to activate or amplify Ca^2+^ signals [29,30,31].

The neuronal plasma membrane, equipped with channels, receptors, and pumps, is responsible for mediating electrical signal propagation, while the ER membrane plays a crucial role in facilitating both passive and active calcium-based signaling, primarily along the dendritic length and soma, from the periphery toward the cell center [12]. Similar to the attenuation seen in the passive propagation of electrical potentials, a small influx of calcium through ER receptors or VGCCs results in the passive diffusion of calcium ions to adjacent regions [10,32]. This diffusion process contributes to localized calcium signaling, modulating various cellular functions.

## 3. Inositol 1,4,5-Trisphosphate Receptors and Ryanodine Receptors

Ca^2+^ can be released from the ER through IP_3_Rs and RyRs. G-protein-coupled receptors are linked to IP_3_ production through the Gq protein, which activates phospholipase C β (PLC β). IP_3_ then stimulates Ca^2+^ release from the ER via IP_3_Rs [33,34]. IP_3_-mediated calcium release in the soma in the first place and dendrites is typically triggered by the action of neurotransmitters, such as glutamate [35]. This release regulates physiological and pathological processes within neurons, as evidenced by studies [36,37,38,39,40].

RyRs are activated by an increase in cytosolic Ca^2+^ concentration. Calcium entry through VGCCs can stimulate RyRs via a mechanism widely known as “calcium-induced calcium release” (CICR). Several factors regulate the activity of RyRs, including Ca^2+^ (a primary channel agonist), Mg^2+^ (an endogenous antagonist), ATP, calmodulin, and various protein kinases and phosphatases. Notably, the cellular redox state influences RyR activation by Ca^2+^ through CICR. Under reducing conditions, this activation is inhibited, while RyR oxidation facilitates channel activation by Ca^2+^ [41]. Ryanodine receptors participate in calcium signals during prolonged pyramidal neurons stimulation, at least at the [Ca^2+^]_i_ levels reached in these stimuli [14]. Recent studies have also established a link between RyR-mediated Ca^2+^ release and nuclear Ca^2+^ signaling [40].

Both InsP_3_Rs and RyRs have a bell-shaped dependence on cytosolic calcium levels, with low calcium levels acting as a coactivator for these receptors, while higher levels suppress their activity [42,43,44,45,46]. During active calcium signaling, CICR amplifies ER calcium release, creating a positive feedback loop that results in significant elevations in cytosolic calcium. The released calcium can diffuse to nearby receptors, enhancing calcium flux and leading to regenerative calcium release from the ER stores, which can propagate as calcium waves across the cell. However, once cytosolic calcium levels rise steeply, they act as inhibitors to ER receptors, shutting down further calcium release from the ER [4,47,48].

The rodent hippocampus predominantly expresses the RyR2 and RyR3 isoforms, while RyR1 is present at lower levels [49,50]. Recent research utilizing specific antibodies that exclusively recognize the rat RyR1 isoform, and not RyR2 or RyR3, has revealed that the adult male rat hippocampus lacks RyR1, with RyR1 mRNA levels detectable only at the threshold of qPCR sensitivity [49,51,52]. Further investigations confirmed the presence of native RyR2 channels within the soma and dendrites of primary rat hippocampal neurons. These channels are absent from dendritic spines, which express only the RyR3 isoform [53]. Notably, RyR3 channels amplify the activity that induces Ca^2+^ signals in postsynaptic dendritic spines [54]. Both RyR and IP_3_R channels may interact to enhance and propagate Ca^2+^ signals within neurons, supporting signal transduction processes [15,40].

Numerous studies highlight the importance of RyR-mediated Ca^2+^ release in hippocampal learning and memory. Specifically, selective downregulation of RyR2 or RyR3, but not RyR1, has been shown to temporarily and reversibly impair performance in memory tasks, such as passive avoidance tests in mice [55]. RyR2 is particularly critical, as its downregulation in rats leads to deficits in spatial memory formation, and mutations in RyR2 negatively affect hippocampal learning and memory [52,56]. Furthermore, training in spatial memory, object location, or fear conditioning protocols has been shown to significantly increase hippocampal levels of the RyR2, RyR3, and IP_3_R1 proteins [49,51,57].

Aging is often associated with deficits in synaptic transmission and plasticity, which are linked to impairments in learning and memory [58,59]. A key factor in these age-related changes is the increased oxidative stress in aged neurons, which leads to excessive activation of RyR-mediated Ca^2+^ release—a process sensitive to redox conditions [59,60]. Notably, inhibiting the RyR channels or using antioxidant agents has been shown to significantly reduce the prolonged Ca^2+^-dependent slow afterhyperpolarization phase observed in aged hippocampal neurons [60]. Moreover, the RyR antagonist dantrolene has been found to mitigate age-related spatial memory deficits, indicating that controlling excessive RyR-mediated Ca^2+^ release may have neuroprotective effects on aging neurons [61].

In hippocampal neurons, the potassium voltage-gated channel Kv2.1 facilitates the clustering of the somatic, L-type calcium channel Cav1.2, which modifies their kinetics and enhances their activity [62]. Abundant studies have established a link between VGCCs, calcium-sensitive potassium channels, and RyRs [63]. Ionotropic glutamate receptors, including N-methyl-D-aspartate (NMDA) and α-amino-3-hydroxy-5-methyl-4-isoxazolepropionic acid receptors (AMPA) receptors, play a significant role in signaling at endoplasmic reticulum–plasma membrane (ER-PM) junctions [64].

Depletion of Ca^2+^ from the ER initiates a cascade leading to the activation of store-operated Ca^2+^ entry (SOCE) across the plasma membrane [65,66]. SOCE is triggered when the reduction in the ER Ca^2+^ level causes Ca^2+^ to dissociate from the luminal Ca^2+^-binding sites on the ER protein stromal interaction molecule 1 (STIM1). STIM1 then undergoes a conformational change, exposing cytosolic domains that bind to the Ca^2+^ channel Orai1 on the plasma membrane, resulting in the Orai1 channel opening and a Ca^2+^ influx into the cell [67,68,69]. Furthermore, IP_3_Rs can modulate SOCE in neurons. The binding of IP_3_ to these receptors facilitates the association between STIM1 and Orai1 independently of their Ca^2+^-releasing function. This enhanced interaction between STIM1 and Orai1 occurs in neuronal cells with depleted ER Ca^2+^ stores, promoting SOCE activation [69].

## 4. Local Ca^2+^ Events

As mentioned above, the IP3R and RyR channels expressed in the ER also generate Ca^2+^ signals in response to neuronal activity. The ER forms a network that permeates the entire neuron volume, providing intracellular Ca^2+^ release in dendrites, soma, and presynaptic boutons.

At the local level, the activity of the RyRs is recorded as calcium spikes, which are considered as changes in calcium concentration during its release from the SR receptors due to the simultaneous opening of several RyR channels, commonly known as “sparks,” which were first identified in cardiac myocytes [70] and in frog skeletal muscle fibers [71]. These sparks were quickly recognized as fundamental units of the larger, regenerative Ca^2+^ release essential for muscle contraction, resulting from the opening of RyR clusters in the ER through local CICR. Similarly, IP_3_R-mediated Ca^2+^ release events were first reported in Xenopus laevis Oocytes [72] and HeLa cells [73]. Often referred to as “puffs,” they require not only Ca^2+^ but also IP_3_ in the cytoplasm to activate IP_3_Rs. When a cluster of RyRs is located near a cluster of IP_3_Rs, the Ca^2+^ released by the RyRs can activate additional IP_3_Rs in the nearby cluster more effectively than the baseline intracellular Ca^2+^ concentration with the presence of IP_3_ [74].

In neurons, localized events with properties similar to sparks were first identified in differentiated PC12 cells and hippocampal neurons [75]. Ca^2+^ sparks occurred more frequently at branch points, a property that was later observed in dendrites of intact pyramidal neurons in slices [8]. Localized calcium events, called “syntillas,” were observed in the presynaptic terminals of hypothalamic neurons [76]. Local Ca^2+^ release events in dendrites have been observed in developing chick retinal ganglion cells and hippocampal pyramidal neurons, stabilizing dendrite outgrowth and correlating with synapse formation. These events, activated by cholinergic and GABAergic signaling, have different durations depending on their proximity to the synapses [77]. The role of sparks in neurons is less well understood than in muscles. The presence of extensive subsurface endoplasmic reticulum suggests that neuronal sparks are instrumental in mediating bidirectional communication between the plasma membrane and the ER, which is localized within the cell.

In pyramidal neuron dendrites, Ca^2+^ sparks, similar to those in myocytes and other cells, are mediated by RyRs, occur spontaneously, and have rapid rise times (<10 ms). They also decay quickly, aided by a combination of diffusion and membrane pump activity [78]. Localized Ca^2+^ release events in presynaptic terminals and cell bodies have been observed in various neurons, with possible roles in synaptic transmission and plasticity. These RyR-mediated events have a spatial extent of 5–10 μm and a duration of 0.2–2.0 s [4]. Some evidence in cultured hippocampal neurons suggests that ER calcium release via RyRs is essential for the downregulation of A-type potassium channels [79].

At the local level, RyRs and IP_3_Rs are responsible for local calcium events, which are transient fluctuations in calcium concentration that result from the coordinated opening of multiple RyRs or IP_3_Rs. By understanding the role of sparks and puffs mediated by RyRs and IP_3_Rs in both normal and pathological conditions, we can gain insight into functional changes in Ca^2+^ signaling at the cellular and subcellular levels.

## 5. Ca^2+^ Waves

Calcium release from the ER in neurons is essential for both short- and long-term synaptic plasticity, as well as for sustaining calcium waves that propagate within and between cells [4,80,81,82,83]. These Ca^2+^ waves are a form of regenerative calcium signaling, where elevated cytoplasmic Ca^2+^ levels trigger further Ca^2+^ release through a positive feedback mechanism known as CICR, creating a non-linear, cooperative process. This signaling can occur via IP_3_Rs or by the activation of RyRs. IP_3_R-mediated Ca^2+^ waves were initially characterized in non-neuronal systems, such as Xenopus laevis Oocytes and HeLa cells, where they contribute to various cellular responses [73,84,85]. Similarly, RyR-mediated Ca^2+^ waves have been well-documented in cardiac myocytes, where they play a critical role in coordinating muscle contraction [86].

Both IP_3_Rs and RyRs produce localized and transient Ca^2+^ signals from specific ER microdomains, which lead to distinct neuronal responses [87]. These two types of channels also interact both spatially and temporally to generate larger, more sustained, and propagating Ca^2+^ signals [74,75]. These broader signals can, in turn, amplify extracellular Ca^2+^ entry, initiating regenerative Ca^2+^ release [88,89,90].

An increase in peripheral intracellular calcium concentration propagates centripetally as a calcium wave via CICR, driven by a diffusion–reaction mechanism [91,92,93]. In the pyramidal neurons, the calcium response to action potentials appeared comparable to that observed in atrial myocytes [14,92]. The complexity of this process in hippocampal neurons is increased by the presence of not only RyRs but also a high density of IP_3_ receptors, requiring not only calcium activation of RyRs in the central ER but also IP_3_ receptors, achieved by diffusion of IP_3_ from the cell periphery into the central ER to bind to non-submembrane regions of the ER [94].

ER-mediated Ca^2+^ waves in hippocampal neurons are significant for signaling pathways involved in learning, memory, and other essential neuronal functions. Synaptically driven IP_3_-mediated Ca^2+^ waves are found in pyramidal neurons within the hippocampus, cortex, and rodents’ amygdala, underscoring their widespread role across various brain regions [95,96,97]. In hippocampal and cortical pyramidal neurons, Ca^2+^ waves typically originate in the primary apical dendrite, with occasional propagation into the soma and slight extensions (approximately 10–20 μm) into the oblique and basal dendrites. These waves facilitate the spread of calcium signals across the neuron, contributing to diverse physiological responses and synaptic plasticity [98,99].

Ca^2+^ waves have been proposed as a mechanism for transmitting signals from dendritic synapses to the nucleus, where elevated [Ca^2+^]_i_ could activate specific genes or transcription factors involved in synaptic plasticity [90,95,100,101]. However, the localized nature of IP_3_ release, typically confined to regions near the activated synapse, suggests that these waves are unlikely to reach the soma and relay synaptic messages under normal conditions, except in specific, rare cases [4].

Figure 2 shows a schematic representation of a neuron with the major components of the organelles involved in calcium signaling and how the endoplasmic reticulum system is distributed throughout the neuron, including the axon, dendrites, and spines. The peak [Ca^2+^]_i_ amplitude of ER-mediated Ca^2+^ waves can exceed 5 μM when measured with non-buffering, low-affinity Ca^2+^ indicators, which is considerably higher than the 0.15–0.3 μM range seen in a VGCC-mediated Ca^2+^ influx from back-propagating action potential (bAP) in the same dendritic regions [98,102,103]. Furthermore, the duration of synaptically evoked Ca^2+^ waves is typically prolonged (0.5–1.5 s) compared to the brief (0.02–0.1 s) Ca^2+^ transients induced by ligand-gated or spike-triggered Ca^2+^ signals [104]. These Ca^2+^ waves propagate at an average velocity of ~100 μm/s when recorded with low-affinity indicators, whereas bAP-evoked Ca^2+^ signals can traverse this distance in approximately 0.5 ms [98,105].

Recent studies have explored how ER geometry and Ca^2+^ release contribute to Ca^2+^ waves in neurons. It has been found that a minimum density of RyRs is essential for initiating Ca^2+^ waves in dendrites, similar to observations in cardiac myocytes [106]. Additionally, dendrites with a smaller ER network require a higher density of RyRs than those with a more extensive ER network. This suggests that the capacity to trigger a Ca^2+^ wave is proportional to the rate of Ca^2+^ release from the ER, with larger ER structures facilitating wave initiation at lower RyR densities [107].

Ca^2+^ signaling in hippocampal pyramidal neurons has been demonstrated to be RyR-dependent, exhibiting similar calcium propagation dynamics from the periphery to the center, as observed in the atrial myocardium [14,108]. It is crucial to acknowledge the potential for a refractory mechanism of inhibition of Ca^2+^ release from the ER in hippocampal pyramidal neurons, resulting from alterations in the kinetics of release restitution. This could potentially impair the activity of the nerve cell, which may manifest in specific changes in the activity of ryanodine receptors. Such disruptions in RyR activity may contribute to the pathophysiology of various neurodegenerative diseases, including Parkinsonism, Alzheimer’s disease, and conditions associated with hypoxia.

## 6. Dendrites, Spines and Calcium Signaling

Neuronal dendrites are crucial for signal integration, neural computation, plasticity, and the overall adaptability of neuronal structures [12]. Within the dendritic tree, small membranous protrusions known as dendritic spines serve as the primary postsynaptic sites for glutamatergic synapses. As mentioned above, in neurons, the ER is found in the soma and also observed at postsynaptic sites, where it is estimated to be present in approximately 40% of dendritic spines [109]. The ER within dendritic spines, referred to as the spine apparatus (SA), plays a role in synaptic plasticity in certain contexts, though its precise function remains unclear [110,111,112]. The SA and associated Ca^2+^ release in spines may enhance synaptic plasticity by extending the duration of the Ca^2+^ signal within the spine [113] or by facilitating its propagation into the dendrite [107,114]. A key feature of signaling in dendritic spines is the ability to convert brief electrical signals into localized, synapse-specific biochemical responses, delivering synapse-specific Ca^2+^ signals [11].

Specific synaptic activation can lead to the recruitment of intracellular Ca^2+^ stores through a Ca^2+^ influx from the external environment via voltage- and ligand-gated ion channels. This recruitment subsequently triggers the release of Ca^2+^ from the ER via intracellular Ca^2+^ release channels, including IP_3_Rs and RyRs, within spines [54,110,115,116,117] and dendrites [90]. RyRs can establish specialized calcium signaling nanodomains within individual spines. The clustering of VGCCs in the presynaptic active zone is crucial for their well-documented role in neurotransmitter release [118]. The influx of Ca^2+^ into spines through the plasma membrane is facilitated by NMDA receptors and VGCCs [74]. VGCCs located in these dendritic structures facilitate bAPs [105,119,120], enabling dendrites to generate local dendritic spikes. When action potentials are fired, most spines experience global back-propagating action potential-induced calcium transients. These bAPs are electrochemically linked to calcium release from the ER through RyRs. Functionally, calcium release mediated by RyRs in these nanodomains generates a novel form of calcium transient plasticity, which acts as a spine-specific storage mechanism for neuronal patterns of suprathreshold activity [3].

Synaptic plasticity, including long-term depression and long-term potentiation, modulates neuronal synapses and is linked to memory formation and loss. Calcium signaling plays a crucial role in both processes. Brief trains of glutamatergic [121] or cholinergic [101] synaptic stimulation (or both [122]) in hippocampal slices generate localized Ca^2+^ waves that are usually confined to the dendrites. With strong and sustained stimulation, enough IP_3_ can cause waves that could spread into the soma. In that case, an intense Ca^2+^ release response is often generated in the cell body. Ca^2+^ waves cause a large and long-lasting [Ca^2+^]_i_ increase in the dendrites. It is reasonable to suggest that these waves could induce plasticity [95,101,121].

## 7. Mitochondrial and Endo-Lysosome Calcium Signaling

The ER also formed networks around mitochondria and made contact with endosomes, multivesicular bodies, and lysosomes [20]. An increase in cytoplasmic calcium concentration promotes Ca^2+^ binding to cytoplasmic buffers of varying Ca^2+^ affinities, and if sufficiently high due to the proximity of mitochondria with the ER. Also, Ca^2+^ released from the ER could directly transfer to nearby mitochondria, influencing Ca^2+^ oscillations [123,124]. This process involves Ca^2+^ accumulation in the mitochondrial matrix by its translocation across the outer and inner mitochondrial membranes. The outer membrane is permeable to ions and small solutes, and voltage-gated anion channels mediate Ca^2+^ uptake. The inner membrane, however, is a tight barrier impermeable to ions. The mitochondrial calcium uniporter (MCU) mediates mitochondrial Ca^2+^ uptake mainly through membrane potentials [125]. The MCU enables rapid accumulation of Ca^2+^ in the mitochondrial matrix in response to cytosolic Ca^2+^ concentration increases caused by specific agonists [126]. Excessive mitochondrial Ca^2+^ loading through the MCU can lead to the opening of the mitochondrial permeability transition pore, ultimately causing mitochondrial damage and triggering apoptosis or necrosis [127,128].

Mitochondrial Ca^2+^ plays a crucial role in regulating cellular energy metabolism. Mitochondrial Ca^2+^ waves were observed following 200 action potentials at frequencies of 40 Hz or 20 Hz, but not at lower frequencies or with fewer action potentials. The use of inhibitors targeting the mitochondrial calcium uniporter and oxidative phosphorylation significantly suppressed these mitochondrial Ca^2+^ waves. Notably, blocking AMPA and NMDA receptors did not affect the propagation of mitochondrial Ca^2+^ waves. Mitochondrial Ca^2+^ waves were present in hippocampal neurons during electrical stimulation and occurred when cytosolic Ca^2+^ concentration was elevated, helping to prevent the overloading of cytosolic or mitochondrial Ca^2+^ through lateral diffusion [129].

Mitochondria play an important role in shaping the Ca^2+^ signal released from the ER. The mitochondria assist with the recovery phase by rapidly sequestering Ca^2+^ and then later returning it to the ER. Another organelle—endo-lysosome—also stores a decent amount of calcium. Endosomes and lysosomes (endo-lysosomes, ELs) are membrane-bound acidic organelles that play a key role in cell survival and death and are involved in endocytosis, recycling, and degradation of extracellular and intracellular material [130,131]. ELs contain many biologically important substances, including divalent cations and also calcium, with a lumenal concentration estimated to be about 500 μM, levels approaching those found in the ER [16,132,133].

Ca^2+^ uptake in the ELs occurs via a mechanism that is incompletely characterized; it requires the H^+^ gradient generated by the V-type H^+^-ATPase pump together with Na^+^/H^+^, Ca^2+^/H^+^, and perhaps Na^+^/Ca^2+^ exchangers whose identity is still controversial [134,135,136,137]. Nicotinic acid adenine dinucleotide phosphate (NAADP) is the most potent Ca^2+^ mobilizing second messenger [134,138,139]. NAADP produced in response to agonist stimulation activates Ca^2+^ release from the ELs. It could trigger Ca^2+^ release from the ER via IP_3_Rs in the presence of IP_3_ or via RyRs through a CICR mechanism [134].

Communication with the ER, such as Ca^2+^ released from endo-lysosomes, can increase the ER calcium load [140]. Changes in the pH of endo-lysosomes can release calcium from them, which can initiate mechanisms such as vesicular fusion of late endosomes and lysosomes [141,142], and calcium release from acidic calcium stores can also depolarize plasma membranes, evoke calcium-dependent currents, and stimulate a calcium influx across plasma membranes [143].

EL Ca^2+^ stores have been identified in several cell models from different species, from invertebrates to mammals, and have also been found in neurons [134]. Calcium release from this acidic organelle store contributes to calcium-dependent processes of fundamental physiological importance for neurons, including neurotransmitter release, membrane excitability, neurite outgrowth, synaptic remodeling, and cell viability. Pathologically, disruption of the EL structure and/or function has been reported in a variety of neurodegenerative diseases, including Alzheimer’s disease, HIV-1-associated neurocognitive disease, Parkinson’s disease, and various forms of brain cancer, such as glioblastoma multiforme (for more information, see reviews [132,143]). The main driver in these disorders is dysregulation of intracellular calcium, which is increased or upregulated.

## 8. Conclusions

Neuronal Ca^2+^ signaling represents a sophisticated and meticulously calibrated system that encompasses a multitude of pathways and cellular compartments. The endoplasmic reticulum, plasma membrane, extensive dendritic system, functional mitochondrial fission, and interaction of these cellular elements contribute to the unique dynamics of Ca^2+^ signals, which are not only influenced by IP_3_ and ryanodine receptors but also by diverse channels, receptors, and pumps of the neuronal plasma membrane that facilitate the transmission of modulated electrical signals. Calcium release from endo-lysosomes can increase the ER calcium load or activate RyRs by CICR. An understanding of these mechanisms not only deepens our knowledge of fundamental neural processes but also has important implications for the treatment of pathologies associated with disturbances of calcium homeostasis in the cell.

Future research should delineate the precise mechanisms by which ER, mitochondrial, and endo-lysosomal calcium release influence cellular homeostasis and dysfunction. The main driver of the final stage of pathological processes in neurological diseases (e.g., Alzheimer’s, Parkinson’s, and epilepsy) is the uncontrolled elevation of calcium concentration, which could be released or extruded from the organelles. Modulation of receptors and channels could reduce unbalanced calcium homeostasis. The identification of novel modulators of RyRs and IP_3_Rs has the potential to open new avenues for therapeutic intervention in diseases characterized by calcium dysregulation. Thus, from a clinical perspective, targeting RyRs- or IP_3_Rs-mediated Ca^2+^ release may be a promising therapeutic strategy to counteract Ca^2+^ signaling dysregulation associated with neurological diseases.

Another area of future research is to investigate the roles of RyRs and IP_3_Rs and the spatial organization of the ER in the firing of action potentials and their propagation into neuronal dendritic trees. These processes encompass both elementary calcium release events and global calcium signaling. Combining experimental data with computer modeling of these processes will allow a deeper understanding of the mechanisms of neural network operation.

## Figures and Tables

**Figure 1 biomolecules-14-01617-f001:**
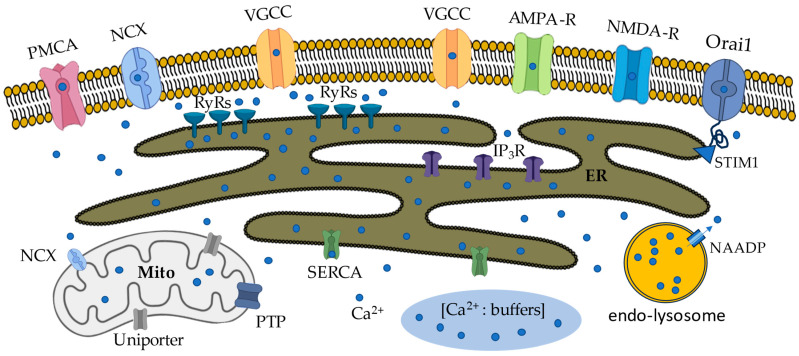
Key channels and receptors involved in Ca^2+^ signaling in hippocampal neurons. The plasma membrane contains voltage-gated Ca^2+^ calcium channels (VGCC) and ionotropic glutamate receptors (N-methyl-D-aspartate, NMDA, or α-amino-3-hydroxy-5-methyl-4-isoxazolepropionic acid, AMPA receptors), and loss of Ca^2+^ from the ER also activates STIM1, which then binds to Orai1 at ER-PM junctions to initiate store-operated Ca^2+^ entry (SOCE), which allow Ca^2+^ entry into neuronal cells. The plasma membrane Na^+^/Ca^2+^ exchanger (NCX) and Ca^2+^ ATPase (PMCA) regulate the free calcium concentration inside the neurons. The Ca^2+^ signal is amplified by Ca^2+^ release from the endoplasmic reticulum (ER) via ryanodine receptors (RyRs) and inositol 1,4,5-trisphosphate receptors (IP_3_Rs) or removal by sarcoplasmic/endoplasmic reticulum Ca^2+^-ATPase (SERCA). In addition, calcium buffers and mitochondria are involved in signal filtering. Mitochondria include NCX, calcium uniporter, and permeability transition pore (PTP) involved in mitochondrial calcium signaling. Nicotinic acid adenine dinucleotide phosphate (NAADP) is produced in response to agonist stimulation that activates Ca^2+^ release from endo-lysosomes.

**Figure 2 biomolecules-14-01617-f002:**
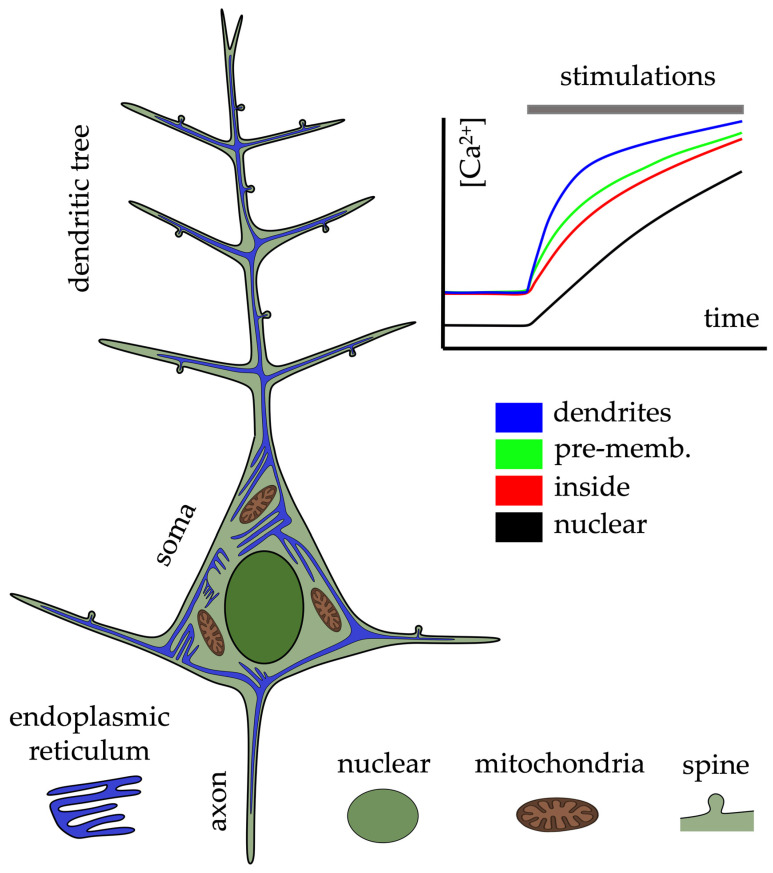
Schematic representation of a pyramidal neuron. Endoplasmic reticulum network through the entire neuron at the axon, soma, dendritic tree, and spines. Focused ion beam–scanning electron microscopy revealed numerous ER-PM contacts in the cell body, with fewer links in dendrites, axons, and spines (see [20] for details). Insets from the top right represent changes in calcium concentration under train pulse electrical stimulation obtained in different regions (dendritic tree, pre-membrane, inside out of the membrane, and nuclear) of hippocampal pyramidal neuron (modified from [14]).

## Data Availability

Not applicable.

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
