# Peer review of "Endoplasmic Reticulum Calcium Signaling in Hippocampal Neurons"

_biomolecules, 2024, doi:10.3390/biom14121617_

Round 1
Reviewer 1 Report
Comments and Suggestions for Authors
The review paper highlights the crucial role of the endoplasmic reticulum (ER) in neuronal calcium signaling, which is essential for maintaining cellular homeostasis and supporting synaptic plasticity. It details the mechanisms of calcium release from the ER via inositol 1,4,5-trisphosphate receptors (IP3Rs) and ryanodine receptors (RyRs), while also discussing the contributions of the ER’s structure, mitochondrial calcium uptake, and voltage-gated calcium channels in fine-tuning calcium signals critical for neuronal function. However, the manuscript requires further proofreading, with improvements needed in the logical flow and coherence of the text. Moreover, the conclusions should more explicitly emphasize the potential pathophysiological implications of the mechanisms described. Additional recommendations for enhancing the manuscript are outlined below.
Abstract.
Following sentences required paraphrasing for clarity: “These components collectively influence on formation and propagation of calcium and electrical signals within and outside to other neurons.” - Influence on calcium signal and formation or propagation of electrical stimuli in neuronal network
“By examining the propagation of signals throughout the soma, dendritic tree, plasma membrane, the ER, mitochondria and the associated calcium fluctuations in nerve cells, we can gain insight into the intricate mechanisms underlying Ca²⁺ signaling in neurons.” – by examining calcium fluctuations in nerve cells, and associated mechanisms at the ER, mitochondria, and cytosol, we can gain insights into the mechanistic pathways of nerve cell interactions.
The text strongly required some proofreading by native speakers. I added some examples below and suggest to proofread the whole text for clarity and flow.
Line 26 “Connecting organs and tissues” allowing communication between…
Line 29 “activate voltage-activated channels,” activate voltage-gated channels
Lines 40-41 “the ER membrane… primarily spreading signals along the dendrites”
Line 60 “while also man- 66 aging calcium storage, lipid metabolism”
Line 118 “exhibit a bell-shaped dependence” of what?
The role of STIM1 and ORAI1 channels should be described and depicted in the Figure 1
Line 127 Authors should introduce to the reader why is the main focus was shift specifically to hippocampal neurons
Line 277 “and subsequently through and also through”
The sparks term should be better described in lay terms that reader understand the meaning and characteristics of this physiological evets
“bAP- 254 evoked Ca2+ signals” all abbreviations and terminology should be introduced in the text
It will be nice to add EM photo shown the presence of ER in dendrite and spines
The conclusions should emphasize the broader perspectives and significance of the physiological mechanisms reviewed in the paper. Additionally, the first paragraph should be relocated to the introduction to provide a concise overview of the research topic, setting the stage for the detailed discussion that follows in subsequent sections.
Comments on the Quality of English Language
The text strongly required some proofreading by native speakers.
Author Response
I would like to thank the referee for the comments and a frank assessment toward improving the manuscript.
1. Following sentences required paraphrasing for clarity: “These components collectively influence on formation and propagation of calcium and electrical signals within and outside to other neurons.” - Influence on calcium signal and formation or propagation of electrical stimuli in neuronal network
“By examining the propagation of signals throughout the soma, dendritic tree, plasma membrane, the ER, mitochondria and the associated calcium fluctuations in nerve cells, we can gain insight into the intricate mechanisms underlying Ca²⁺ signaling in neurons.” – by examining calcium fluctuations in nerve cells, and associated mechanisms at the ER, mitochondria, and cytosol, we can gain insights into the mechanistic pathways of nerve cell interactions.
Thank you for pointing this out. I agree with these comments and have changed it in abstract: “These channels and receptors are involved in the generation and transmission of electrical signals and the modulation of calcium concentrations within the neuronal network. By analyzing calcium fluctuations in neurons and the associated calcium handling mechanisms at the ER, mitochondria, endo-lysosome and cytosol, we can gain a deeper understanding of the mechanistic pathways underlying neuronal interactions and information transfer.”
- The text strongly required some proofreading by native speakers. I added some examples below and suggest to proofread the whole text for clarity and flow.
Thank you for your feedback! I have improved the text further based on your comments to ensure better clarity and flow.
Line 26 “Connecting organs and tissues” allowing communication between…
Changed: lines 26-27 “allowing communication between organs to the central nervous system”
Line 29 “activate voltage-activated channels,” activate voltage-gated channels
Agree, new_line 39 “voltage-gated channels”, changed.
Lines 40-41 “the ER membrane… primarily spreading signals along the dendrites”
New_lines 55-56, changed to “The ER membrane also contributes to the propagation of calcium-based signals, primarily along the dendritic length and soma”
Line 60 “while also man- Line 66 aging calcium storage, lipid metabolism”
New_lines 114-115, changed to: “…and coordinating key processes such as protein and lipid synthesis, folding and storage…”
Line 118 “exhibit a bell-shaped dependence” of what?
New_lines 155-157, the text was modified:” have a bell-shaped dependence on cytosolic calcium levels, with low calcium levels acting as a coactivator for these receptors, while higher levels suppress their activity [42-46].”
The role of STIM1 and ORAI1 channels should be described and depicted in the Figure 1
Agree, was added to figure 1 and the legend.
Line 127 Authors should introduce to the reader why is the main focus was shift specifically to hippocampal neurons
Thanks for pointing this out. I have changed the title of the article to: "Endoplasmic reticulum calcium signaling in hippocampal neurons". I also added the importance of the hippocampus and hippocampal neurons in the introduction (please see new_lines 27-36). It was my mistake to use all neurons and not make comparisons between them, it could be a new additional paper covering differences in ER, RyRs, IP3Rs and pre-membrane channels in neurons of different structures of the CNS.
Line 277 “and subsequently through and also through”
Changed, new_lines 370-373: The text was modified: “This process involves Ca2+ accumulation in the mitochondrial matrix occurs by its translocation across the outer and inner mitochondrial membranes. The outer membrane is permeable to ions and small solutes, and voltage-gated anion channels mediate Ca2+ uptake.”
The sparks term should be better described in lay terms that reader understand the meaning and characteristics of this physiological evets
I agree with you. I rewrite the sparks’ part with your recommendations, please see: new lines 210-250.
“bAP- 254 evoked Ca2+ signals” all abbreviations and terminology should be introduced in the text
The bAP was introduced in the text, new_line 309.
It will be nice to add EM photo shown the presence of ER in dendrite and spines
I add necessary information into figure 2 legend, please see text, new_lines 293-295: “Focused ion-beam scanning electron microscopy revealed numerous ER-PM contacts in the cell body, with fewer links in dendrites, axons, and spines (see [20] for details)”.
The conclusions should emphasize the broader perspectives and significance of the physiological mechanisms reviewed in the paper. Additionally, the first paragraph should be relocated to the introduction to provide a concise overview of the research topic, setting the stage for the detailed discussion that follows in subsequent sections.
I totally agree with you, part of conclusion was replaced into introduction, new_lines 51-62.
Please check new version of conclusion, new_lenes 422-435.
Reviewer 2 Report
Comments and Suggestions for Authors
This review revisits the intricate mechanisms of calcium (Ca²⁺) signaling in neurons, emphasizing its central role in cellular homeostasis and neural function. It delves deeper into the endoplasmic reticulum's (ER) critical role in calcium dynamics, highlighting its structural integration of sheets and tubules that facilitate calcium-based signaling propagation. It further explores the interplay between voltage-gated and ionotropic glutamate receptors in the plasma membrane, as well as intracellular channels such as ryanodine and inositol-1,4,5-trisphosphate receptors within the ER.
Special attention is given to the propagation of signals through the neuronal soma, dendritic tree, plasma membrane, ER, and mitochondria, examining how these interconnected systems mediate calcium fluctuations and electrical signal integration. The revision aims to provide a more comprehensive analysis of these processes, addressing their implications for neuronal communication, plasticity, and overall cellular health. By refining the discussion on these dynamics, the review seeks to deepen our understanding of Ca²⁺ signaling's pivotal role in neurons and its broader physiological and pathological contexts.
I have minor comments:
1) line 29 "Action potentials activate..." I would suggest to stay more general such as "Depolarizing stimuli activate..."
Line 32: instead of "or control in excitable cells many physiological pathways" I would suggest "or control many ohysiological pathways in excitable cells"
line 32-35: this list is misleading since the processes that are listed do not necessarily belong to excitable cells, such as fertilization, or at least are present also in other cell type (non- excitable). Please consider to change.
Line 56: remove "and" before transcription
Line 67: consider to substitute "storage, lipid metabolism" with "storage and lipid metabolism"
IN figure 1 consider to substiture SOCE with Orai
Legend Fig. 1 line 73 remove "and" before "ionotropic"
Line 116: the symbol [Ca2+i] has never been introduced before, thus pleae define it
line 245: please insert the word "representation" after "schematic"
Line 250: after "back propagating action potential" introduce the acronym b-AP that is used few lines later
Line 263, I suggest to remove "on" before RyRs at the end of the line
Line 319: Use directelty the acronym bAP that was already introduced
Lines 328-330: the word "generate/generated" apprears three times. Please fins synonyms
Lines 33-353: this part is not related to a Conclusion of the review; it looks like a summary of all already said. I suggest to remove it and start the paragraph with "In conclusion,..."
Please note that the reference number 74 does not appear in the manuscript.
Author Response
I wish to thank the reviewer for the thorough reading of my manuscript and the sharing of his/her expertise and valuable insight into the problem at hand.
Line 29 "Action potentials activate..." I would suggest to stay more general such as "Depolarizing stimuli activate..."
New_line 39, It was changed on “voltage-gated channels”.
Line 32: instead of "or control in excitable cells many physiological pathways" I would suggest "or control many ohysiological pathways in excitable cells"
New_lines 42-43, the text was modified:”…controls many physiological pathways in excitable cells…”
Line 32-35: this list is misleading since the processes that are listed do not necessarily belong to excitable cells, such as fertilization, or at least are present also in other cell type (non- excitable). Please consider to change.
I agree, I removed fertilization, new_lines 42-45.
Line 56: remove "and" before transcription
I agree and changed.
Line 67: consider to substitute "storage, lipid metabolism" with "storage and lipid metabolism"
New_lines 113-115, the text was modified:”…The ER is a vital cellular organelle that plays a central role in maintaining homeostasis by regulating calcium levels and coordinating key processes such as protein and lipid synthesis, folding and storage [19].
IN figure 1 consider to substiture SOCE with Orai
I added in the figure 1 and legend about orai1 and stim1 based on the recommendation of review 1
Legend Fig. 1 line 73 remove "and" before "ionotropic"
New_lines 66, “and” was removed.
Line 116: the symbol [Ca2+i] has never been introduced before, thus pleae define it
The symbol [Ca2+i] was introduced in new_line 52
Line 245: please insert the word "representation" after "schematic"
I added schematic into the text, please see new_line 299
Line 250: after "back propagating action potential" introduce the acronym b-AP that is used few lines later
Changed with your recommendation, new_lines 304
Line 263, I suggest to remove "on" before RyRs at the end of the line
“on” was removed, please check on new_line 318
Line 319: Use directelty the acronym bAP that was already introduced
In new_line 348 changed on bAPs
Lines 328-330: the word "generate/generated" apprears three times. Please fins synonyms
The text was modified, new_lines 359-362:” With strong and sustained stimulation, enough IP3 can cause waves that could spread into the soma. In that case, an intense Ca2+ release response is often generated in the cell body. Ca2+ waves cause a large and long-lasting [Ca2+]i increase in the dendrites”
Lines 33-353: this part is not related to a Conclusion of the review; it looks like a summary of all already said. I suggest to remove it and start the paragraph with "In conclusion,..."
I removed and placed into introduction part, new_lines 51-62
Please note that the reference number 74 does not appear in the manuscript.
Thank you, it was updated.
Reviewer 3 Report
Comments and Suggestions for Authors
The review manuscript entitled "Endoplasmic Reticulum Calcium Signaling in Neurons" discusses the role of ER and ER membrane receptors in maintaining intracellular calcium homeostasis in excitable cells. The topic is of major interest, especially in the field of neurophysiology and cellular signaling. The manuscript is written and is clear and easy to follow. However, some missing information needs to be added and discussed to enhance the comprehensiveness of the context
· Minor issue:
Lines 56-57: Provide related references supporting the role of calcium signaling in neurological disorders.
· Major issues:
1) While the review provides valuable insights into the neuronal ER calcium, it does not adequately consider the heterogeneity of neurons in CNS.
Neurons differ in their structure, morphology, and expression of ion channels, both on the plasma membrane and within endogenous membranes. These differences raise concerns about generalizing findings related to ER calcium signaling across all CNS neurons.
Thus, the authors should clarify the specific neuronal subtypes under discussion and highlight the distinct calcium signaling properties of these subtypes. expalining that ER calcium signaling mechanisms may vary between neuronal populations will provide a more nuanced and accurate perspective.
2) Endolysosomes, an additional reservoir of calcium ions besides ER, play a major role in maintaining intracellular calcium homeostasis. studies have demonstrated the sensitivity of IP3 receptors to localized calcium released by lysosomal ion channels, by the mechanism of CICR. Thus, I recommend adding a dedicated section to discuss the interaction between endolysosomal calcium stores and ER calcium signaling, discussing its role in cytosolic calcium dynamics and its potential implications for neuronal firing.
Author Response
I would like to thank the reviewer for reading of my manuscript and for helpful comments.
Minor issue:
Lines 56-57: Provide related references supporting the role of calcium signaling in neurological disorders.
Necessary referenced were provided, please check new_line 106-107
Major issues:
1) While the review provides valuable insights into the neuronal ER calcium, it does not adequately consider the heterogeneity of neurons in CNS.
Neurons differ in their structure, morphology, and expression of ion channels, both on the plasma membrane and within endogenous membranes. These differences raise concerns about generalizing findings related to ER calcium signaling across all CNS neurons.
Thus, the authors should clarify the specific neuronal subtypes under discussion and highlight the distinct calcium signaling properties of these subtypes. expalining that ER calcium signaling mechanisms may vary between neuronal populations will provide a more nuanced and accurate perspective.
Thank you very much for your thoughtful and constructive comments. I understand and appreciate your concern about the possible diversity of calcium signaling in neurons in different parts of the CNS. There are differences in the organization of the ER network and the expression of receptor types and channels in different structures, and this could possibly be the subject of a separate paper. To address this issue, I have changed the title of the paper to emphasize hippocampal neurons. This narrower scope reflects the main neuronal population discussed in our review and reduces the risk of overgeneralization.
New title: “Endoplasmic reticulum calcium signaling in hippocampal neurons.” Also was added importance of the hippocampus and hippocampal neurons into introduction part (pleases see new_lines 27-36).
2) Endolysosomes, an additional reservoir of calcium ions besides ER, play a major role in maintaining intracellular calcium homeostasis. studies have demonstrated the sensitivity of IP3 receptors to localized calcium released by lysosomal ion channels, by the mechanism of CICR. Thus, I recommend adding a dedicated section to discuss the interaction between endolysosomal calcium stores and ER calcium signaling, discussing its role in cytosolic calcium dynamics and its potential implications for neuronal firing.
I added part about endo-lysosomes into text, please see new_lines 364 – 420. Also added endo-lysosome into figure 1.
Round 2
Reviewer 1 Report
Comments and Suggestions for Authors
The author has addressed all my comments; however, the conclusion section lacks a discussion on the perspectives and advancements in the current field. It also fails to address specific questions related to pathological processes and the clinical significance of the described mechanisms and studies.
Author Response
Thank you for your constructive feedback. I have made the necessary improvements to the conclusion section, including a discussion on the perspectives and advancements in the field. I hope these revisions meet your expectations, and I appreciate your valuable input in enhancing the manuscript.
Conclusions
Neuronal Ca2+ signaling represents a sophisticated and meticulously calibrated system that encompasses a multitude of pathways and cellular compartments. The endoplasmic reticulum, plasma membrane, extensive dendritic system, functional mitochondrial fission, and interaction of these cellular elements contribute to the unique dynamics of Ca2+ signals, which are not only influenced by IP3 and ryanodine receptors but also by diverse channels, receptors, and pumps of the neuronal plasma membrane that facilitate the transmission of modulated electrical signals. Calcium release from endo-lysosomes can increase ER calcium load or activate RyRs by CICR. An understanding of these mechanisms not only deepens our knowledge of fundamental neural processes but also has important implications for the treatment of pathologies associated with disturbances of calcium homeostasis in the cell.
Future research should delineate the precise mechanisms by which ER, mitochondrial, and endo-lysosomal calcium release influence cellular homeostasis and dysfunction. The main driver of the final stage of pathological processes in neurological diseases (Alzheimer's, Parkinson's, epilepsy e.g.) is the uncontrolled elevation of calcium concentration, which could be released or extruded from the organelles. Modulation of receptors and channels could reduce unbalanced calcium homeostasis. The identification of novel modulators of RyRs and IP₃Rs has the potential to open new avenues for therapeutic intervention in diseases characterized by calcium dysregulation. Thus, from a clinical perspective, targeting RyRs or IP₃Rs-mediated Ca2+ release may be a promising therapeutic strategy to counteract Ca2+ signaling dysregulation associated with neurological diseases.
Another area of future research is to investigate the role of RyRs, IP₃Rs, and the spatial organization of the ER in the firing of action potentials and their propagation into neuronal dendritic trees. These processes encompass both elementary calcium release events and global calcium signaling. Combining experimental data with computer modeling of these processes will allow a deeper understanding of the mechanisms of neural network operation.
Reviewer 3 Report
Comments and Suggestions for Authors
No further comments, all the raised issues are addressed accurately.
Author Response
Thank you for your positive feedback. I glad to hear that all the raised issues have been addressed to your satisfaction. I appreciate your time and consideration in reviewing my work.
Round 3
Reviewer 1 Report
Comments and Suggestions for Authors
All comments have been addressed.